# Bacterial Utilisation of Aliphatic Organics: Is the Dwarf Planet Ceres Habitable?

**DOI:** 10.3390/life12060821

**Published:** 2022-05-31

**Authors:** Sahan A. Jayasinghe, Fraser Kennedy, Andrew McMinn, Andrew Martin

**Affiliations:** 1Institute for Marine and Antarctic Studies, University of Tasmania, Hobart 7004, Australia; sahan.jayasinghe@utas.edu.au (S.A.J.); f.c.kennedy@utas.edu.au (F.K.); 2School of Biological Sciences, Victoria University of Wellington, Wellington 6012, New Zealand

**Keywords:** Ceres, astrobiology, *Colwellia hornerae*, aliphatic hydrocarbons

## Abstract

The regolith environment and associated organic material on Ceres is analogous to environments that existed on Earth 3–4 billion years ago. This has implications not only for abiogenesis and the theory of transpermia, but it provides context for developing a framework to contrast the limits of Earth’s biosphere with extraterrestrial environments of interest. In this study, substrate utilisation by the ice-associated bacterium *Colwellia hornerae* was examined with respect to three aliphatic organic hydrocarbons that may be present on Ceres: dodecane, isobutyronitrile, and dioctyl-sulphide. Following inoculation into a phyllosilicate regolith spiked with a hydrocarbon (1% or 20% organic concentration wt%), cell density, electron transport activity, oxygen consumption, and the production of ATP, NADPH, and protein in *C. hornerae* was monitored for a period of 32 days. Microbial growth kinetics were correlated with changes in bioavailable carbon, nitrogen, and sulphur. We provide compelling evidence that *C. hornerae* can survive and grow by utilising isobutyronitrile and, in particular, dodecane. Cellular growth, electron transport activity, and oxygen consumption increased significantly in dodecane at 20 wt% compared to only minor growth at 1 wt%. Importantly, the reduction in total carbon, nitrogen, and sulphur observed at 20 wt% is attributed to biotic, rather than abiotic, processes. This study illustrates that short-term bacterial incubation studies using exotic substrates provide a useful indicator of habitability. We suggest that replicating the regolith environment of Ceres warrants further study and that this dwarf planet could be a valid target for future exploratory missions.

## 1. Introduction

Microorganisms on Earth have evolved a range of capabilities to metabolise organic substrates for energy in the absence of more favourable nutrients [1,2,3,4,5]. Aliphatic organic compounds in the form of hydrocarbons are found in a multitude of environments where they are biosynthesised by a taxonomically diverse assemblage of organisms including lichenised fungi, bryophytes, macrophytes, phytoplankton, and bacteria [6,7]. The majority of psychrotolerant hydrocarbon-utilising microbes described to date have been isolated from soils, sediments, and cryconite holes on glaciers [8,9], some of which resemble the localised organic deposits on Ceres and other ice-dominated asteroids [10,11]. In polar regions, bacteria from the genus *Colwellia* can metabolise complex hydrocarbons [8,12]; this genus dominates marine and ice-associated environments and is integral to the recycling of complex organics and food web dynamics [13].

The Dawn spacecraft orbited Ceres between 2015 and 2018, collecting high-resolution images, infrared and elemental spectroscopic data, and gravity measurements [14,15]. Patches of ice were detected at Ceres’s high latitudes, where surface temperatures and solar irradiance levels are lower, implying that water and ice are likely abundant in the subsurface. This was later confirmed by Dawn’s Visible and InfraRed mapping spectrometer (VIR) [15,16]. However, unlike the oceans of Europa and Enceladus, the VIR data suggest that Ceres’ ocean resembles a deep brine layer with an ice content of 30–40 vol%, and rocks, salts, and/or clathrates comprising the remaining 60–70 vol% [15,17,18]. Furthermore, the global surface profile most closely resembles a G-type carbonaceous asteroid [19], predominantly composed of ammonium-bearing and magnesium-bearing phyllosilicates and carbonates [15]. Several bright regions blotch the Cerean surface and appear to be remnants of liquid brines that had percolated to the surface and dried over time [20]. Spectral analysis of the most prominent bright spots (Cerealia Facula) indicate a primary composition of sodium carbonate and ammonium carbonate salts—inorganic compounds that, to date, have only been detected on Earth and Enceladus [21,22]. In addition, carbon in the form of organic matter has also been detected on the surface [23,24]. An absorption band at 3.4 um in VIR’s near-infrared spectra suggests this matter is aliphatic; however, the exact composition is unknown [25]. The total carbon content on the surface is estimated to approach 20 wt%, nearly five times the concentration of chondritic meteorites and other carbonaceous asteroids [19,26,27]. The Cerean surface and subsurface layers appear to be chemically dynamic and rich in carbon, organics, and water—potentially resembling the prebiotic conditions found on Earth 3–4 billion years ago [10,11,27,28]. This environmental complexity and the origin of its constituents is intriguing—carbonaceous chondrites are not only considered the closest analogues to Ceres, but the water and organics within the surface layers may reflect a significant history of regionalised impact events [29,30].

The presence of significant amounts of organics on Ceres’ surface suggests a proto-metabolic environment where microbial metabolism could theoretically occur. These microenvironments may nurture the synthesis of 30–50 monomer RNA units integral to the early development of primitive forms of life [31,32,33]. Clay minerals also have catalytic properties which can cause incidental chemical reactions that influence microenvironments with respect to oxygen flux and pH [34]. Montmorillonite is a clay with a very distinct three-layer structure containing an aluminate layer sandwiched between two silicate layers intwined with magnesium that creates distinct micro-niches [34,35]. In the presence of brines and aqueous solutions, smectite clays such as montmorillonite can undergo “crystalline swelling”, whereby water molecules interact with the clay substrate, reducing its permeability and causing ionic exchanges and substitutions to occur [35]. This alters the chemical equilibrium and generates distinct redox environments that can favour dissimilatory reductive energy-generating pathways [36]. This further exemplifies the possible catalytic and reaction-inducing environment of the Ceres regolith and raises interesting questions about the formation of biogenic precursor molecules.

*Colwellia hornerae*, a psychrophilic and halotolerant facultative anaerobe isolated from Antarctic sea ice, was chosen as an analogous proxy species for this study [37]. Our aim is to determine whether this extremophile can utilise aliphatic organic substrates added to a common phyllosilicate clay that mimics the silicate-rich surface layer of Ceres. Specifically, we examine dodecane, a liquid alkane hydrocarbon commonly found in crude oil and natural seepages; isobutyronitrile, a complex organic nitrile commonly found in meteorites and interstellar media; and dioctyl-sulphide, an organosulphur comparable to the sulphide compounds predicted to exist on Ceres.

## 2. Materials and Methods

### 2.1. Culture Conditions

Cryopreserved vials containing *Colwellia hornerae* (ACAM607) were obtained from the Australian Collection of Antarctic Microbes (ACAM) (Tasmanian Institute of Agriculture, University of Tasmania, Hobart, Australia). These vials were left to thaw for 30 min at 2 °C before being aspirated onto sterile marine agar and incubated at 10 °C. Stock cultures were developed by transferring isolated colonies into 30 mL sterile modified marine broth (MMB) [38]. One litre of MMB contained 35 g sea salts (Sigma Aldrich, St. Louis, MO, USA), 15 g bacteriological peptone (Amyl Media, Dandenong, Australia), and 3 g yeast extract (Sigma Aldrich, St. Louis, MO, USA). Stock cultures were subcultured fortnightly and maintained at 10 °C. The integrity of the cultures was determined using microscopy and cross-referencing with the description of *C. hornerae* provided by Bowman et al. [39].

### 2.2. Experimental Conditions

Cultures of *C. hornerae* were grown in 2 L culture flasks at 2 °C for one month to a cell density of 1 × 10^8^ cells mL^−1^. Before inoculation, the cells were centrifuged and washed three times using a modified nitrogen-free M9 media to remove all traces of MMB [39]. Then, 50 mL (approximately 10^10^ cells) was transferred into a clean sterile vessel containing 250 g of sterilised montmorillonite (Sigma Aldrich, St. Louis, MO, USA) and gently agitated to form a consistent clay substrate. During agitation, volumes of either dodecane, isobutyronitrile, or dioctyl-sulphide were added, corresponding to 1% and 20% organic concentration (wt%). These concentrations were chosen to reflect the total carbon concentration estimated to be on the surface of Ceres (20 wt%) and the minimum concentration likely to elicit a response (1 wt%). Subsamples (15 g) were then extracted and transferred to 15 mL glass tubes (Schott AG, Mainz, Germany) and sealed with a silicone rubber stopper and wrapped in parafilm. Triplicate tubes were prepared for each hydrocarbon, concentration, and time point. Controls were prepared as described above; however, they differed with respect to organic substrate addition; the positive control contained 60 mL of MMB (20 wt%) whereas the negative control did not contain a substrate. Bacterial contamination was assessed using microscopy on three occasions: prior to experimentation (batch working culture), and on days 16 and 32.

### 2.3. Whole-Cell Extraction from Clay Samples

Cell samples were extracted using the modified methods of Calbrix et al. [38] and Bressan et al. [40] prior to performing each assay described below. Briefly, subsamples (5 g) were weighed and transferred into sterile glass centrifuge tubes (Corning Inc., Corning, NY, USA) and diluted with 45 mL of sterilised saline solution (0.85% NaCl). The mixture was then homogenised for 5 min before being centrifuged at 1000× *g* for 5 min at 2 °C to exclude large clay particles. The resulting supernatant was used for cell enumeration, ATP, NADPH/NADP^+^, and protein analysis.

### 2.4. Cell Enumeration, Intracellular Activity, and Growth Analysis

An assessment of intracellular electron transport activity combined with cell counts was performed using epifluorescence microscopy. Specifically, 5-cyano-2,3-ditolyl tetrazolium chloride (CTC), which is a non-fluorescent dye converted into a red-coloured fluorescent formazan in cells undergoing respiration, was added to each sample and incubated in the dark for three hours at 2 °C (50 µL for a final concentration of 5 mM, Cayman Chemicals, Toronto, Canada). Cells were then fixed with glutaraldehyde (0.5% final concentration) and stored at 2 °C. Prior to enumeration, the cells were counter-stained with 4′,6-Diamidino-2-phenyindole hydrochloride (DAPI) (100 µL for a final concentration 0.5 µg mL^−1^, Sigma-Aldrich, St. Louis, MO, USA) and incubated in the dark for 15 min. Counter-stained cells were then vacuum filtered onto a 0.22 µm black polycarbonate membrane (Merck Millipore, Boston, MA, USA) using a 25 mm filtering apparatus (Merck Millipore). The membrane was removed and placed on a glass slide with a drop of SlowFade antifade reagent (Molecular Probes, Eugene, OR, USA) before a coverslip was overlain and sealed with clear nail polish. Cells were visualised under oil immersion at ×1000 magnification on a Nikon ECLIPSE Ci-S upright microscope equipped with a 100 W mercury lamp (Nikon, Tokyo, Japan). A UV excitation filter was used to count DAPI-stained bacteria (365 nm excitation, 400 nm beam splitter, and 400 nm barrier filter), and a blue-light excitation filter for CTC-positive cells (450–490 nm excitation, 505 nm beam splitter, and 520 nm barrier filter). For each sample, 10 fields of view were randomly selected, and counts were performed to determine total cell number (DAPI) and the number of actively respiring cells (CTC-positive). These counts were then extrapolated to account for sample volume and the total number of fields of view at ×1000 magnification. A modified Gompertz equation [41] was used to fit the growth curves for each treatment:(1)$$y\left(t\right)=y_{0}+\left(y_{max}-y_{0}\right)exp\left(-\exp(\mu_{max}\left(t-M\right)\right)\;$$
where *y(t)* is the cell abundance (cells mL^−1^) at time *t*; *y*_0_ is the initial cell abundance (cells mL^−1^); *y_max_* is the population-carrying capacity, µ_max_ is the maximum relative growth rate (day^−1^) at *t = M*; *M* is the time at which the absolute growth rate is maximum (day). The growth parameters (*y*_0_, *y_max_*, *µ_max_*_,_ and *M*) were obtained using the curve-fitting tool in Matlab (Version 2021a, Math Works, Natwick, MA, USA). The 95% confidence intervals were calculated for the growth parameters, and *r*^2^, SSE, and RMSE were calculated for the curve fit. The derived growth parameters were used to calculate the specific growth rate *r_max_* (day^−1^) and lag phase duration λ (days):(2)$$r_{max}=\frac{y_{max}-y_{0}}{e}\times \mu_{max}\;$$
(3)$$\lambda =M-\frac{1}{\mu_{max}}\;$$

### 2.5. Adenosine Triphosphate (ATP) Analysis

The concentration of intracellular adenosine triphosphate (ATP) was measured with an ELISA ATP kit (Thermo Fisher Scientific, Waltham, MA, USA). The extraction protocol was adapted from Mempin et al. [42] and Finch [43]. Briefly, 1 mL of the sample was pipetted into a centrifuge tube, and 500 µL of ice-cold 2 M sulphuric acid (H_2_SO_4_) was added. Tubes were then vortexed for 10 s and left to incubate on ice for 15 min then placed in a refrigerated centrifuge (4 °C) and spun at 8000× *g* for 15 min. The resulting supernatant was transferred into a sterile centrifuge tube with 500 µL of ice-cold pH-neutralising solution (0.8 M KOH and 0.2 M Na_2_CO_3_). Tubes were spun again at 8000× *g* for 15 min at 4 °C before the supernatant was aspirated and stored at −80 °C. Before analysis, samples were thawed, and 10 µL of the extracted ATP was added to a 96-well plate and then mixed with 90 µL of assay reagents in the dark according to the manufacturer’s instructions. Luminescence was measured immediately using a FLUOstar Optima microplate reader (BMG Labtech, Ortenberg, Germany) to determine ATP concentration.

### 2.6. Nicotinamide Adenine Dinucleotide Phosphate (NADP^+^/NADPH) Analysis

Calculation of NADP^+^/NADPH ratios was determined using a fluorometric extraction kit (Abcam, Cambridge, UK). Reagents and samples were prepared in the dark, according to the manufacturer’s instructions. Briefly, subsamples (25 µL) were added to microplate well containing NADPH or NADP^+^ extraction solutions and left to incubate in the dark for 10 min at room temperature. After incubation, additional extraction reagents were added to each well and incubated for a further 60 min. Fluorescence was then measured using a FLUOstar Optima microplate reader (BMG Labtech, Ortenberg, Germany) and the NADP^+^/NADPH ratio was calculated by comparing samples to standards.

### 2.7. Protein Content

Cells were obtained by centrifuging 30 mL subsamples at 7800× *g* for 30 min at 4 °C. Pellets were rinsed once in ice-cold phosphate-buffered saline (PBS), resuspended, and stored at −80 °C. Cell lysis was carried out using a Bioruptor Plus (Diogenode) at high power for 10 min (15 s on, 15 s off sonication cycle) at 4 °C. Following sonication, cells were centrifuged at 15,000× *g* for 15 min at 4 °C. The soluble fraction was carefully separated, and total protein was then measured using a Bradford assay (Thermo Fisher Scientific, Waltham, MA, USA).

### 2.8. Measurement of O_2_ Consumption

A concurrent experiment was performed to examine rates of O_2_ consumption. Samples were prepared in vials as described in Section 2.2 and were kept sealed for the duration of the experiment. Oxygen concentration was determined using oxygen optodes (SP-Pst3-NAU, PreSens GmbH, Regensburg, Germany) fixed to the inner wall of the vials. A 2-point calibration (0% and 100% air saturation) was conducted at the onset of the experiment for each optode. Measurements were recorded via Fibox 4 portable meter as described by the manufacturer (PreSens GmbH, Regensburg, Germany). Oxygen concentration (µmol O_2_ L^−1^ h^−1^) over time was normalised to cell density and compared between days 1 and 32 to derive a relative rate of oxygen consumption.

### 2.9. Elemental Analysis

To examine changes in the clay composition over time, 2 g of each sample was diluted with 3 mL of distilled water and homogenised by benchtop vortex (~2000 rpm) for 5 min. The resulting mixture was transferred to a Ziplock bag and stored at −80 °C. To remove water content without losing N and S as volatiles, samples were subsequently dried for 48 h in a FreeZone 2.5 L benchtop freeze dryer (Labconco, Kansas City, MO, USA) at −56 °C and <0.02 mbar. Once dry, samples were crushed into a fine powder and transferred into 1.5 mL vials. The vials were stored in air-locked containers with activated desiccant under vacuum. To measure the change in total carbon, nitrogen, and sulphur, samples were analysed using a Thermo Finnigan EA 1112 Series Flash Elemental Analyser after being weighed in a Sartorius Microbalance SE2 w. Following blank correction, the total elemental content data were normalised to cell density.

### 2.10. Statistical Analysis

A two-way mixed analysis of variance (ANOVA) was used to assess differences among treatments with respect to cell density, metabolic activity, and the production of ATP, NADPH, and protein by *C. hornerae*. Differences in r_max_ and O_2_ consumption relative to substrate and concentration were assessed using a one-way ANOVA. If the interaction between treatment and time on the respective parameter was statistically significant (*p* < 0.05), a Tukey post hoc pairwise comparison test was conducted. If there was no homogeneity of variance, a Games–Howell pairwise comparison was conducted instead with the Welch Statistic and corrected degrees of freedom values reported. A nonparametric Spearman’s rank-order correlation test was used to measure the strength and direction of the relationship between total elemental content and time for each treatment. A relationship was deemed to have a strong correlation if the Spearman correlation coefficient (*r_s_*) was >±0.5. Two-tailed significance levels are reported. Data are presented as the mean ± standard error unless otherwise stated. All statistical analysis was performed using SPSS (Version 27, IBM, Armonk, NY, USA).

## 3. Results

### 3.1. Cell Density and Growth

Cell density was highest in the positive control (substrate MMB, 1.8 ± 0.1 × 10^8^ cells g^−1^ on day 32) and lowest in the negative control (no substrate, 1.0 ± 0.03 × 10^8^ cells g^−1^ day 32, Figure 1). With respect to the growth of *C. hornerae* in the three hydrocarbons, a significant interaction was observed between incubation time and substrate concentration (*F* = 2.06, *p* < 0.05). In the dodecane treatment, cell densities were significantly lower (*F* = 2.22, *p* < 0.05) on day 32 in the 1 wt% (1.1 ± 0.1 × 10^8^ cells g^−1^) compared to 20 wt% substrate concentration (1.6 ± 0.1 × 10^8^ cells g^−1^, Figure 1a). In the isobutyronitrile, an increase in cell density was observed during the experiment, but there was no statistically significant difference between the two concentrations, with a final cell density in the 1 wt% isobutyronitrile of 1.2 ± 0.1 × 10^8^ cells g^−1^ and 1.58 ± 0.1 × 10^8^ cells g^−1^ in the 20 wt% (*F* = 3.2, *p =* 0.958, Figure 1b). In the dioctyl-sulphide, there was no significant difference over time with respect to cell density (*F* = 1.95, *p* = 0.653, Figure 1c). These data were fitted to a modified Gompertz model to examine the growth mechanics of *C. hornerae*. Clear exponential and stationary growth phases were only evident in the positive control (Figure 1, Table 1). A lineal increase in growth was observed during the first 4 days of the experiment in all but one of the treatments (Figure 1); only the 20 wt% isobutyronitrile treatment displayed a lag phase, with significant growth not occurring until after day 4. Although the relative growth rate had declined in all substrates at 20 wt% concentration by day 32, a clearly defined stationary phase was not evident in any of the hydrocarbon treatments.

### 3.2. Physiological Parameters

The proportion of cells with an active electron transport chain (CTC-positive) varied over time (*F* = 8.71, *p* < 0.05). On day 1, activity was similar across all treatments, with only 5–10% of the cells reducing CTC. By day 32, the percentage of CTC-positive cells had doubled across all treatments (Figure 2a–d). In the positive control, activity peaked at 53 ± 7% on day 8, whereas in the negative control the peak was 41 ± 7% on day 16. In the hydrocarbon treatments, activity was highest on day 8 in the 20 wt% isobutyronitrile (53 ± 2%) and secondarily in the 1 wt% concentration (33 ± 3%). The proportion of CTC-positive cells was less in the dodecane treatment and remained relatively constant between days 1 and 8 before increasing on day 16 to 14 ± 2% and 26 ± 3% in the 1 wt% and 20 wt% concentrations, respectively (Figure 2a). In the dioctyl-sulphide, CTC-positive cells gradually increased by day 16 with 21 ± 2% and 17 ± 2% in the 1 wt% and 20 wt%, respectively (Figure 2c).

The intracellular concentration of ATP across all treatments was highest during the first four days of the experiment and then declined over time (Figure 2e–h). The ATP concentration was significantly higher in the hydrocarbon substrates at 1 wt% compared to 20 wt% (*F* = 9.62, *p* < 0.05, Figure 2e–h). Over the 32-day incubation period, ATP concentrations declined in the dodecane treatment by 63% and 74% in the 1 wt% and 20 wt% concentrations, respectively; in the isobutyronitrile by 57% and 86%, respectively; and in the dioctyl-sulphide by 75% and 64%, respectively. In the control treatments, ATP concentrations declined by 80% and 74% in the negative control and positive controls, respectively.

The NADPH/NADP^+^ ratio was significantly higher in the hydrocarbon substrates at a concentration of 20 wt% compared to 1 wt% (*F* = 19.4, *p* < 0.05, Figure 2i–l). In the dodecane, the NADPH/NADP^+^ ratio was 3 to 25 times higher in the 20 wt% concentration compared to the 1 wt% (Figure 2i). In the isobutyronitrile, the NADPH/NADP^+^ ratio was 3 to 16 times higher in the 20 wt% concentration compared to the 1 wt% (Figure 2j). In the dioctyl-sulphide, the NADPH/NADP^+^ ratio was 1 to 6 times higher in the 20 wt% concentration compared to the 1 wt% (Figure 2k). In the controls, a higher NADPH/NADP^+^ ratio was observed in the negative control compared to the positive control throughout the experiment (Figure 2l).

Protein content per cell decreased over time in all treatments, with the highest protein concentrations observed within the first four days (Figure 2m–p). In the dodecane, protein content was significantly higher at all time points at the concentration of 1 wt% compared to 20 wt% (*F* = 9.26, *p* < 0.05) (Figure 2m). Final protein content on day 32 was 4.8 ± 0.6 pg cell^−1^ and 2.3 ± 0.2 pg cell^−1^ in the 1 wt% dodecane and 20 wt% dodecane, respectively. In the isobutyronitrile, protein content did not differ over time at either concentration (*F* = 2.81, *p* = 0.99). Final protein content on day 32 was 4.8 ± 0.4 pg cell^−1^ and 3.4 ± 0.2 pg cell^−1^ in the 1 wt% and 20 wt% concentrations, respectively. In the dioctyl-sulphide, protein content was higher in the 20 wt% compared to the 1 wt% at all time points except for day 16 (Figure 2o), but protein content did not vary significantly over time between the two concentrations (*F* = 6.1, *p* = 0.07). Final protein content on day 32 was 4.7 ± 0.4 pg cell^−1^ and 5.6 ± 0.6 pg cell^−1^ in the 1 wt% and 20 wt% concentrations, respectively. In the controls, protein content was significantly higher at all time points in the negative control compared to the positive control (*F* = 1.44, *p* < 0.05). The highest protein content in the negative control was 7.1 ± 0.8 pg cell^−1^ which occurred on day 1, whilst the highest protein content in the positive control was 4.5 ± 0.6 pg cell^−1^ and occurred on day 2. Final protein content on day 32 was 3.9 ± 0.2 pg cell^−1^ and 2.7 ± 0.2 pg cell^−1^ in the negative control and positive control, respectively.

### 3.3. Oxygen (O_2_) Consumption

A decline in oxygen was observed in all experimental treatments (Figure 3). Although there was no statistically significant difference among the substrates at 1 wt% concentration, O_2_ consumption in 20 wt% dioctyl-sulphide (78.6 ± 3.1%) was significantly higher compared to the 20 wt% dodecane (60.3 ± 1.8%) and 20 wt% isobutyronitrile (64.0 ± 3.1%) (*F* = 13.84, *p* < 0.05). The highest oxygen consumption was observed in the positive control while change in the negative control was negligible (8.1 ± 5.2%) and non-significant.

### 3.4. Elemental Analysis

A statistically significant decline in total carbon over time was observed in the 20 wt% dodecane (*r_s_*(90) = −0.59, *p* < 0.05), 20 wt% dioctyl sulphide (*r_s_*(90) = −0.71, *p* < 0.05), and the positive control (*r_s_*(90) = −0.63, *p* < 0.05), whilst a moderate decline was observed in the 20 wt% isobutyronitrile (*r_s_*(90) = −0.49, *p* < 0.05) (Figure 4a–d). A moderate decline in total carbon was observed in 1 wt% dodecane (*r_s_*(90) = −0.49, *p* < 0.05), 1 wt% isobutyronitrile (*r_s_*(90) = −0.25, *p* < 0.05), and the negative control (*r_s_*(90) = −0.38, *p* < 0.05). A weak but significant increase in total carbon was observed in the 1 wt% dioctyl- sulphide (*r_s_*(90) = 0.26, *p* < 0.05). A strong and significant decrease in total nitrogen over time was observed in the 1 wt% isobutyronitrile (*r_s_*(90) = −0.53, *p* < 0.05), 20 wt% isobutyronitrile (*r_s_*(90) = −0.50, *p* < 0.05), and the positive control (*r_s_*(90) = −0.90, *p* < 0.05). Conversely, no significant change was observed in the negative control (*r_s_*(90) = 0.14, *p* = 0.20) (Figure 4e,g). In the dioctyl-sulphide, a moderate decrease (*r_s_*(90) = −0.48, *p* < 0.05) in total sulphur over time only occurred at 20 wt%, while a significant increase was observed at 1 wt% (*r_s_*(90) = 0.58, *p* < 0.05) (Figure 4f).

## 4. Discussion

Although potentially hostile to life in general, Ceres may harbour biologically permissive domains associated with a subsurface ocean, icy shell, and surface regolith. Because Ceres lacks a substantial atmosphere and is likely exposed to the full range of incident charged-particle radiation, from low-energy solar wind ions (~1 keV kinetic energy) to ultra-high-energy galactic cosmic rays (MeV to TeV energies) [44], any form of subsurface ecology remains highly speculative. However, recent particle interaction modelling by Nordheim et al. [44] shows that the destruction of amino acids on Ceres would exceed a timescale of 100 My at a depth of 10 cm regardless of assumed surface composition, and approaches timescales of 1 Gy at depths of ~1 m or greater. Because this near-surface environment is thought to contain aliphatic hydrocarbons, we examined the response of an ice-associated Antarctic bacterium, *C. hornerae*, to dodecane, isobutyronitrile, and dioctyl-sulphide. While these data are considered preliminary, given that the experimental design does not account for multiple physiological stressors, our study provides a foundation for determining whether *C. hornerae* could grow on Ceres at depths of >10 cm.

Given the challenging growth substrates, we opted to conduct this experiment with a high inoculum of cells to maximise the detection of physiological change. Importantly, the presence of oxygen within the Ceres regolith remains ambiguous; although *C. hornerae* is a facultative anaerobe (capable of survival and growth in both aerobic and anaerobic conditions), our results are considered within the context of short-term aerobic metabolism.

### 4.1. High Aliphatic Substrate Concentration Governs Utilisation Potential

The ability of bacteria to effectively utilise and biodegrade hydrocarbons is dependent on the substrate concentration [3,45,46]. In this study, the extremophile *C. hornerae* displayed increased growth, metabolic activity, oxygen consumption, and intracellular redox potential (NADPH/NADP^+^ ratio) when the availability of dodecane and isobutyronitrile was high (20 wt%). This corresponded with a reduction in total carbon, nitrogen, and sulphur. In contrast, only minor growth was observed in substrates at a lower concentration (1 wt%). In the negative control, negligible growth and elemental loss occurred, indicating that growth and elemental degradation can be attributed to biotic, rather than abiotic, processes.

The ATP measurements were conducted to evaluate the physiological response to possible toxicity and stress caused by the hydrocarbons. The ATP was highest during the onset of the experiment (4 to 8 days) and was elevated at 1 wt% substrate concentrations compared to 20 wt%. This finding is comparable to previous studies using respiratory activity to assess chemical toxicity in bacteria [47,48]. The amount and rate of ATP synthesis in bacteria has been shown to decrease with increasing hydrocarbon concentration [47]. This may be due to the ability of hydrocarbons to alter and disrupt the peptidoglycan outer-cell membrane and impact the release of hydrolytic enzymes that are essential in the breakdown of organic substrates [48]. Furthermore, the mechanisms that enable the metabolic breakdown of hydrocarbons compounds are complex and highly energetically demanding [49]. A potential strategy to lower the cost may relate to the release of other compounds, such as extracellular polymeric substances (EPS) and omega-3 rich fatty acids [37]. Both are known to act as biosurfactants that lower the surface tension and increase the solubility of recalcitrant hydrocarbons. This release may increase hydrocarbon bioavailability and utilisation potential without disrupting cellular membrane functionality [50]. Interestingly, Bælum et al. [51]) suggest that the ability of *Colwelliae* spp. to produce EPS conferred a competitive advantage in the high hydrocarbon conditions during the Deepwater Horizon oil spill. The observed increase in cellular protein in this study may infer a type of adaption to a hydrocarbon-rich environment, and further work that characterises specific protein functionality is warranted.

The oxidation of organic compounds generates reducing equivalents (NADPH) that are essential for the anabolic reactions that can result in the synthesis of ATP [52]. In our study, NADPH/NADP^+^ ratios were significantly higher in the 20 wt% substrates, which is in contrast to observed ATP content. This result is not surprising given that imbalances between NADPH and ATP are commonly observed during high rates of metabolic expenditure [50,52,53,54]. Importantly, NADPH is also involved in the anti-oxidation defence mechanisms that protect the cell from toxic reactive oxygen species (ROS) produced as byproducts of oxygenic pathways [55,56]. The involvement of NADPH in these processes can create bottlenecks in productivity and ATP generation that create a further decoupling between NADPH and ATP concentrations [52].

High aliphatic organic concentrations can be toxic and detrimental to microbial growth [57]. This was a factor at the beginning of the experiment when the proportion of metabolically active cells remained low for up to 8 days in the dodecane and dioctyl-sulphide treatments, and 2–4 days in the isobutyronitrile treatment. This likely indicates that *C. hornerae* struggled to acclimate to the new and relatively challenging environmental conditions. Nevertheless, growth and utilisation occurred after a lag phase of 4–8 days which is relatively quick compared to other bacterial species with latent hydrocarbon utilisation abilities [58,59,60,61,62]. Following a period of acclimation, we show that *C. hornerae* can withstand the toxicity of high hydrocarbon concentrations and does not require intermediate organic byproducts produced by other hydrocarbon biodegrading bacteria to begin aliphatic biodegradation. A similar observation was made by Lofthus et al. [62], who suggested that the resilience and biosurfactant-producing ability enabled members of *Colwellia* spp. to outcompete *Alcanivorax* spp., which are known to almost exclusively utilise aliphatic hydrocarbons for growth.

### 4.2. Dodecane

Dodecane (C12) is a medium-chain alkane (a saturated non-polar compound lacking functional groups), and is liquid at room temperature and atmospheric pressure [1]. Typical aerobic alkane biodegradation requires the use of a multicomponent alkane monooxygenase system for the initial oxidation of the alkane into the corresponding alcohol [1]. However, the alkBAC operon encoding this system is not present in the *C. hornerae* genome (but is present in other *Colwellia* strains) [1,63,64,65]. Instead, we hypothesise that the degradation of dodecane in this study was via an alkyl hydroperoxide system. This alternative alkane biodegradation pathway was first postulated by Singer and Finnerty [66] and observed in *Acinetobacter* sp. [67,68]. The first step of this pathway involves a dioxygenase catalyst that incorporates dioxygen into the terminal methyl group of dodecane to form dodecyl hydroperoxide. This is then oxidised by an alkyl hydroperoxide reductase into dodecanol and lauric acid. The resulting alcohol can be further hydrolysed whilst the fatty acid can be mineralised via β-oxidation and the tricarboxylic acid cycle (TCA) to produce reducing equivalents for respiratory and energy pathways [67]. This alkyl hydroperoxide system could also explain the high NADPH/NADP^+^ ratio which reflects a partial reduction of O2 and the generation of oxygen free radicals. Because anoxia was not observed in the dodecane treatments, it is unlikely that anaerobic activity would have occurred. Interestingly, dodecane utilisation in *C. hornerae* occurred irrespective of access to nitrogen or other nutrients; this contrasts to *Colwellia* sp. RC25, which was only able to biodegrade when N and P elemental concentrations were replete following the Deepwater Horizon spill [51,69]. Of particular interest is the variability observed with respect to intracellular protein, specifically the reduction in total protein at 20 wt% compared to 1 wt%. While difficult to interpret, this finding highlights the need for enhanced experimental resolution, specifically the use of gene and/or protein expression profiling, in future studies.

### 4.3. Isobutyronitrile

Research on the utilisation of isobutyronitrile, a branched and toxic nitrile compound, is limited to few select bacterial strains [70,71]. Isobutyronitrile consists of a carbon atom bound to two methyl (-CH3) groups and a cyano (-CN) group. In heterotrophic bacteria, two differing enzymatic systems are described that act to convert nitriles into their respective carboxylic acids for metabolism. One system involves a single-step process using nitrilase to hydrolyse nitrile into acid and ammonia. The second system uses a two-step process using the metalloenzyme nitrile hydratase to hydrolyse nitrile into an intermediary amide which is further converted into carboxylic acids and ammonium by amidases [72,73]. In *Colwellia* spp., both systems are present, but a similar system has yet to be described in *C. hornerae*. However, a M20/M25/M40 family of metallohydrolases is present in this species with high similarity to a nitrilase in *C. pontecala*, and two genes are present which encode a carbon–nitrogen hydrolase family protein. *Colwellia hornerae* has a high number of amidase genes, suggesting that the two-step approach is most likely used to utilise isobutyronitrile. This hydrocarbon has been detected in interstellar space gas clouds of star-forming regions, meteorites, and comets [74,75,76], and is predicted to be incorporated into carbonaceous asteroids [77]. Isobutyronitrile’s branched structure has led to the notion that it could have played a role in the origin of life as a precursor to amino acids due to its composition as one of the few -CN containing molecules comprised of straight carbon backbones to be detected in the interstellar medium [74]. Interestingly, *C. hornerae* would have had limited exposure to isobutyronitrile in nature, and the existence of isobutyronitrile utilisation pathways could suggest it may have evolved adjacent to, or in conjunction with, amino acid biosynthesis.

### 4.4. Dioctyl-Sulphide

The negative effect of dioctyl-sulphide on *C. hornerae* was significant. Growth limitation in this substrate was not unexpected given that only lithotrophic bacteria and archaea are known to oxidise sulphur-based compounds for energetic metabolism [78]. Additionally, anoxygenic sulphur reduction pathways have not been identified in *Colwellia* spp. [79,80,81]. Sulphur atoms are the most abundant hetero-atom contained within hydrocarbons [82], and only select highly specialised biodegrading bacteria have developed the ability to incorporate this sulphur for anabolic metabolism by splitting the bonds between C–S atoms [83,84]. It is not known if these mechanisms exist in *C. hornerae,* but several genes encoding sulphur transport proteins are found, suggesting that inorganic and oxidised forms could be scavenged for secondary metabolism [13,79]. Sulphur is a vital element for biological processes and is found in all living organisms, most noticeably in the form of cystine and methionine in nucleic and amino acids as thiols in co-enzymes, and iron–sulphur proteins involved in electron transport [85,86,87]. Sulphur was previously predicted to exist on Ceres in the form of sulphides (and potentially sulphates) based on UV observations by the Hubble Space Telescope [88], but it was not detected on Cerean surface by the recent passing Dawn spacecraft [89]. Bu et al. [90] speculate that this was due to limitations of the VIR spectrometer rather than the absence of sulphides on the surface.

## 5. Conclusions

This study illustrated the short-term viability of an ice-associated bacterial analogue exposed to conditions likely present on Ceres. While this data reaffirms Ceres as a valid target for future exploratory missions, the findings are considered preliminary, and highlight the need to undertake experimental work across multiple physicochemical extremes to accurately represent extraterrestrial environments of interest. Psychrophiles such as *C. hornerae* have historically been considered within the context of cold-active enzymes; our results highlight enzymatic activity associated with hydrocarbon utilisation, and this warrants further work to characterise these enzymes and the extent of their bioremediation capabilities. Finally, this study contributes to our understanding of whether the presence or absence of organic compounds that sustain microbial metabolism provide a useful indicator for characterising extraterrestrial habitability.

## Figures and Tables

**Figure 1 life-12-00821-f001:**
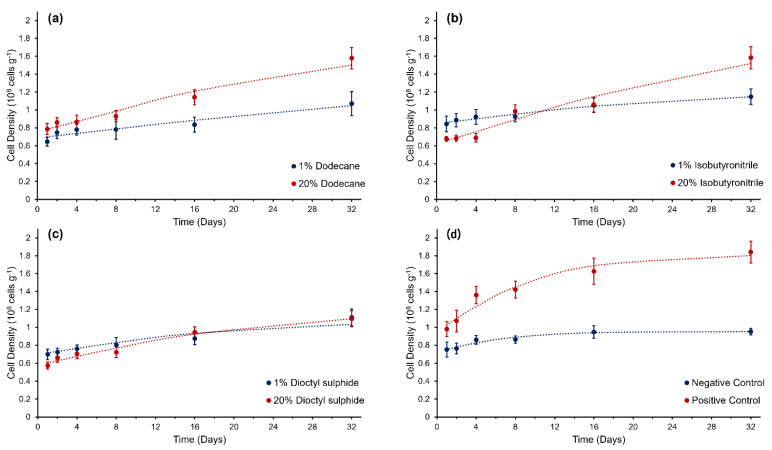
Growth curves of *Colwellia hornerae* grown under Ceres regolith conditions with three organic aliphatic substrates: (**a**) dodecane, (**b**) isobutyronitrile, and (**c**) dioctyl−sulphide; and (**d**) MMB (positive control) and no organic substrate (negative control). Modified Gompertz growth models (dotted lines) have been fitted to growth curves. Error bars represent the standard error of the mean.

**Figure 2 life-12-00821-f002:**
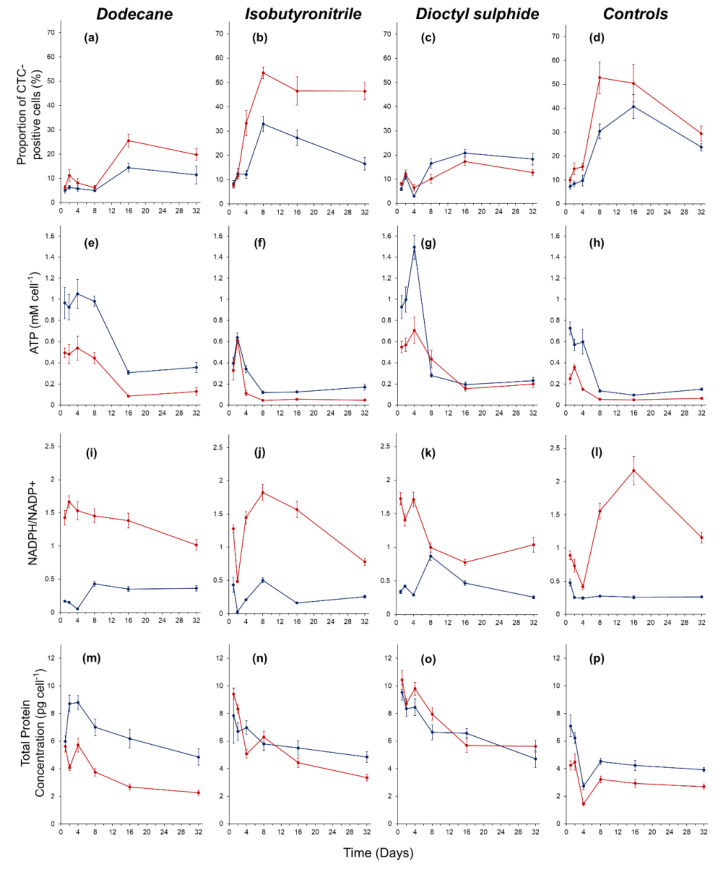
Physiological response of *Colwellia hornerae* under Ceres regolith conditions when supplied with organic substrates: dodecane (**a**,**e**,**i**,**m**), isobutyronitrile (**b**,**f**,**j**,**n**), dioctyl-sulphide (**c**,**g**,**k**,**o**), and controls (**d**,**h**,**l**,**p**). (**a**–**d**) Proportion of CTC-positive (i.e., metabolically active) cells over time: (**e**–**h**) ATP production (mM cell^−1^), (**i**–**l**) NADPH/NADP^+^ ratio, and (**m**–**p**) total protein concentration (pg cell^−1^). Blue lines represent 1 wt% treatment (in aliphatic substrates) and no growth substrate (negative control). Red lines represent 20 wt% treatment (in aliphatic substrates) and MMB (positive control). Error bars represent standard error of the mean.

**Figure 3 life-12-00821-f003:**
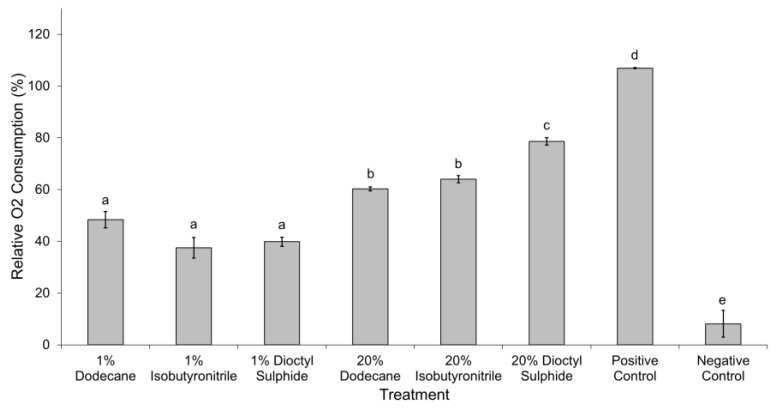
Relative amount of total oxygen consumed by *Colwellia hornerae* over the 32-day experimental period when grown under Ceres regolith conditions in different organic aliphatic substrates (dodecane, isobutyronitrile, and dioctyl-sulphide) at two concentrations (1 wt% and 20 wt%), and MMB (positive control) and no organic substrates (negative control). Values are averages of *n* = 15; errors are standard errors of the mean. Values marked with the same letters are not significantly different from the corresponding factor (Games–Howell multi-comparison test; *p* > 0.05).

**Figure 4 life-12-00821-f004:**
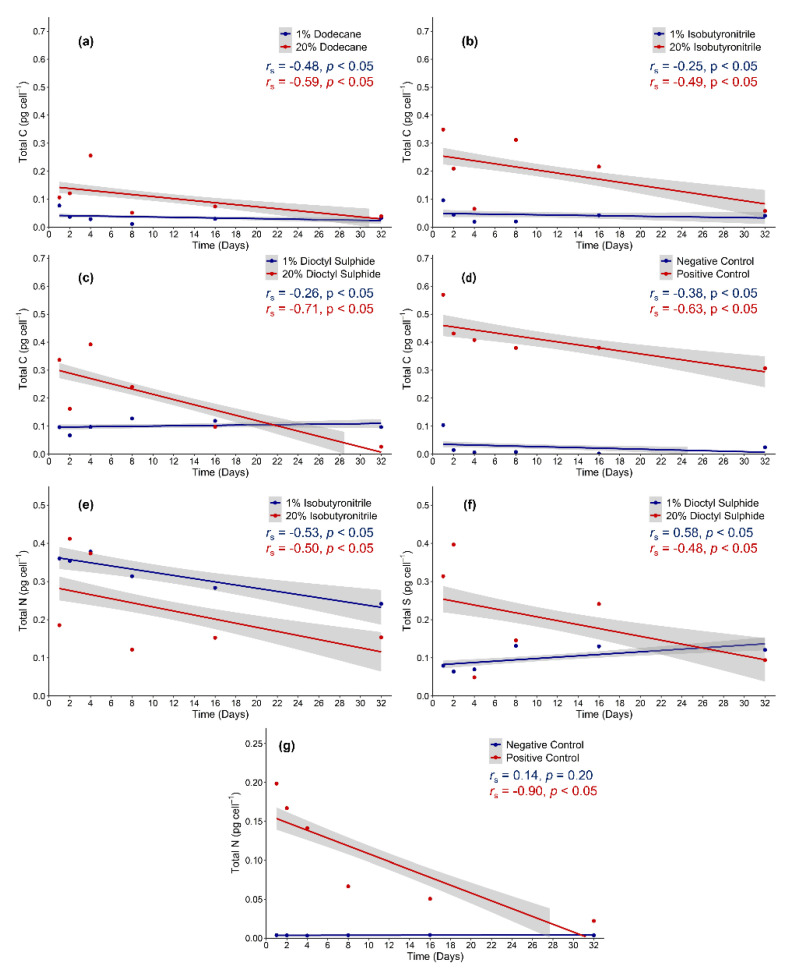
Spearman correlations of total carbon, nitrogen, and sulphur concentrations in *Colwellia hornerae* under Ceres regolith conditions. Total carbon concentrations (pg cell^−1^) in (**a**) dodecane, (**b**) isobutyronitrile, (**c**) dioctyl-sulphide, and (**d**) control substrates. Total nitrogen concentrations (pg cell^−1^) in (**e**) isobutyronitrile and (**g**) control substrates. Total sulphur concentrations (pg cell^−1^) in (**f**) dioctyl-sulphide.

**Table 1 life-12-00821-t001:** Growth parameters and 95% confidence limits with coefficients of determination (R^2^), SSE, and RMSE of fit obtained with modified Gompertz growth model for *Colwellia hornerae* incubated under Ceres regolith conditions with three aliphatic organic substrates and experimental controls.

Substrate		*Y*_0_ (*n* × 10^8^ cells g^−1^)	*Y_max_* (*n* × 10^8^ cells g^−1^)	RGR_max_ (day^−1^)	M	*r_max_* (day^−1^)	Λ (days)	R^2^	SSE	RMSE
Dodecane	1%	0.4 (−30.7, 31.5)	1.3 (−17.9, 20.6)	0.04 (−1.9, 2.0)	5.4 (−1036, 1047)	0.012	3.3 × 10^−10^	0.91	0.009	0.067
	20%	0.6 (−5.8, 6.9)	1.7 (−1.3, 4.7)	0.1 (−0.1, 0.7)	7.6 (−104, 119)	0.029	1.7 × 10^−3^	0.97	0.016	0.089
Isobutyronitrile	1%	0.7 (−3.6, 5.1)	1.2 (0.3, 2.1)	0.2 (−0.5, 0.7)	3.2 (−185.4, 191.9)	0.013	1.9 × 10^−5^	0.97	0.002	0.030
	20%	0.3 (−15.4, 15.9)	1.8 (−5.5, 9.1)	0.1 (−0.8, 0.9)	6.5 (−222.2, 235.2)	0.034	2.3 × 10^−5^	0.96	0.026	0.114
Dioctyl Sulphide	1%	0.6 (−6.2, 7.4)	1.1 (−0.04, 2.2)	0.1 (−1.0, 1.2)	3.5 (−245.2, 252.3)	0.016	4.2 × 10^−4^	0.93	0.010	0.071
	20%	0.38 (−5.0, 5.7)	1.2 (0.04, 2.3)	0.1 (−0.5, 0.6)	4.1 (−130.8, 138.9)	0.023	2.1 × 10^−4^	0.97	0.005	0.052
Control	Negative	0.6 (0.451, 0.738)	1.0 (0.9, 1.1)	0.2 (−0.02, 0.4)	1.1 × 10^−11^ (fixed at bound)	0.027	7.3 × 10^−3^	0.94	0.002	0.026
	Positive	0.5 (−16.7, 17.7)	1.8 (1.2, 2.5)	0.2 (−0.7, 0.9)	1.1 × 10^−7^ (−138.3, 138.3)	0.073	1.0 × 10^−3^	0.95	0.026	0.113

## Data Availability

The data presented in this study is available on request from the corresponding author.

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
