# Peer review of "Bacterial Utilisation of Aliphatic Organics: Is the Dwarf Planet Ceres Habitable?"

_life, 2022, doi:10.3390/life12060821_

Round 1

Reviewer 1 Report

The study by Jayasinghe et al. is interesting and presented in a logical way.

I have no specific comments.

Reviewer 2 Report

Dear Authors,

This is great manuscript, but I have some suggestions, questions and corrections. All are marked in attached file as track changes.

Kind Regards,

Reviewer 3 Report

The article presents a study about the bacterial utilisation of aliphatic hydrocarbons and their relationship with habitability in Ceres. 

The article is well-written and organised, and the results are very interesting.

My field of expertise is Chemistry, so I am not qualified to assess the validity of the bacterial methodology described.

But, regarding the chemical part of this article, it is unclear why the authors used 1% and 20% of the aliphatic hydrocarbons.  Why these specific concentrations?

The results show that there is oxygen consumption by the bacteria. Are there suggestions that there is oxygen in Ceres? Where is this oxygen coming from?

The use of the aliphatic hydrocarbons by the bacteria is catalysed by any enzymes? Are these enzymes identified?

The protein content of the cells decreased during the 32 days of this experiment. Can you explain these results? If the total nitrogen content from the clay decreases and the protein content of the cells also decreases, is there a possibility that the bacteria are producing gaseous nitrogen compounds? This should be better explained in the text.

Reviewer 4 Report

Comments for The Authors.

I’ve found the manuscript very interesting and my recommendation is to accept  with minor revisions. My 2 remarks are listed below:

  1. What was the reason to start experiments with as high microbial numbers? It seems to me that there were too many microbial cells per 1 g (not per 1 ml; The Authors extracted cells from 5 g subsamples, please kindly correct the results) of clay substrate and this is why The Authors did not record any substantial microbial growth. This is also why part of cells were virtually dormant (CTC vs. DAPI, no increase in protein content, even in positive control). Perhaps starting the experiments
  2. How did the Authors overcome the problem of counting microbial cells when dodecane content was as high as 20%? I would expect much higher error bars in comparison with controls or 1%.

Reviewer 5 Report

   Dear authors,

   I consider that your approach is interesting. Certainly Ceres can be an excellent target to grow bacteria, but the current version of your manuscript seems quite naïve. For example, I'd like to see in your paper a clear description of the radiative environment of Ceres, and how it can affect your organisms (please describe where they should be grown: surface, sub-surface, special compartments, etc...). Your paper should cite the flux of solar and cosmic rays on Ceres, and the way you propose to paliate the effects of this sterilizing radiation on your selected locations. You should clearly state if it is viable at this point, at least as a future experiment to be done by a lander, for example.

      On the other hand, I found that your introduction lacks of relevant papers that should be cited. A good summary of our knowledge on Ceres is in a recent review paper (Shi et al., 2021), like this one for organics (Prettyman T. et al. (2022). In addition, most of the water should be in Ceres as clay minerals and oxides, and several review papers are of interest in this context (Trigo-Rodríguez et al., 2019). Most of the organics and water in the regolith might be implanted by the carbonaceous chondrite flux of meteoroids reaching Ceres surface at hypervelocity over the eons. In consequence, the scenario could be far more complexe.

     Consequently, I think that your paper could be considered for publication once these points can be improved.

       References

       Prettyman T. et al. (2022) Carbon and organic matter in Ceres. In Vesta and Ceres. Insights from the Dawn Mission for the Origin of the Solar System, by Simone Marchi, Carol A. Raymond, and Christopher T., ISBN: 978-1-108-47973-8. Cambridge, UK: Cambridge University Press, 2022, p. 121.

       Shi, X. et al.  (2021) GAUSS (Genesis of Asteroids and Evolution of the Solar System)- A Sample Return Mission to Ceres, Experimental Astronomy, Open Access.

       Trigo-Rodríguez, J.M., Rimola, A., Tanbakouei, S., Cabedo-Soto, V., and Lee, M. R. (2019) Accretion of water in carbonaceous chondrites: current evidence and implications for the delivery of water to early Earth, Space Science Reviews 215:18, 27 pp.

Round 2

Reviewer 5 Report

Dear authors

I found that you properly revised the issues raised for the previous version of the manuscript. Now it reads great, thanks for considering my suggestions.

Congratulations